# Two Versatile Pencil Graphite–Polymer Sensor Electrodes Coupled with Potentiometry and Potentiometric Titration Methods: Profiling Determinations of Vitamin V in Tablets and Urine Samples

**DOI:** 10.3390/s22239128

**Published:** 2022-11-24

**Authors:** Abeer M. E. Hassan, Mohamed A. El Hamd, Mahmoud H. El-Maghrabey, Wael A. Mahdi, Sultan Alshehri, Hany A. Batakoushy

**Affiliations:** 1Department of Chemistry, Faculty of Pharmacy, 6 October University, 6th of October City 12585, Egypt; 2Department of Pharmaceutical Sciences, College of Pharmacy, Shaqra University, Shaqra 11961, Saudi Arabia; 3Department of Pharmaceutical Analytical Chemistry, Faculty of Pharmacy, South Valley University, Qena 83523, Egypt; 4Department of Pharmaceutical Analytical Chemistry, Faculty of Pharmacy, Mansoura University, Mansoura 35516, Egypt; 5Department of Pharmaceutics, College of Pharmacy, King Saud University, Riyadh 11451, Saudi Arabia; 6Department of Pharmaceutical Analytical Chemistry, Faculty of Pharmacy, Menoufia University, Shebin Elkom 32511, Egypt

**Keywords:** sildenafil citrate (vitamin V), potentiometry, potentiometric titration, pencil-fabricated graphite sensor electrode, ion-pair complexation

## Abstract

Herein, we developed a new pencil graphite ion-selective electrode strategy for the broadly used erectile dysfunction medication, sildenafil citrate (SC, vitamin V), for its automated potentiometry and potentiometric titration profiling in marketed tablets and human urine samples. The method was based on ion-pair complexation between SC and sodium tetraphenylborate (Na-TPB) or phosphotungstic acid (PTA), embedded into a pencil-fabricated graphite sensor electrode coated with poly(vinyl chloride, PVC) matrix, which is pre-plasticized with two different pre-studied plasticizers. The modern fabricated electrodes have a proven fast near-Nernstian response for SC over the concentration range of 1.0 × 10^−6^ to 1.0 × 10^−2^ and 1.0 × 10^−5^ to 1.0 × 10^−2^ M, with LODs of 6.5 × 10^−7^ and 5.5 × 10^−6^ over a pH 3–6 for (SC-TPB)- and (SC-PTA)-based membrane sensors, of O-nitrophenyl octyl ether (O-NPOE) and dioctyl phthalate (DOP), respectively. The selectivity coefficients for different interferents, including many inorganic cations, sugars, and/or nitrogenous compounds, were tested and confirmed. Applications of the proposed method were conducted on the determination of SC in its tablets and urine samples under the proper conditions. The percent recovery values were compared with those obtained by an official method and showed an RSD ≤ 0.3% (*n* = 5).

## 1. Introduction

Viagra is an anti-impotence drug with an active ingredient, sildenafil citrate (SC, vitamin V), and its own chemical nomenclature is 1-[4-ethoxy-3-(6,7-dihydro-1-methyl-7-oxo-3-propyl-1H-pyrazolo-[4,3-d] pyrimidin-5-yl) phenyl sulphonyl]-4-methylpiperazine citrate, Figure 1 [1,2]. Physiologically, the erection of the penis is caused by the release of nitric oxide (NO) in the corpus cavernosum, which follows sexual stimulation. Then NO activates a guanylate cyclase that increases the levels of cyclic guanosine monophosphate (cGMP), potentates smooth muscle relaxation in the corpus cavernosum, and is accompanied by the inflow of blood [3]. However, SC has no direct relaxant effect on the isolated human corpus cavernosum but potentiates the effect of NO, after inhibiting phosphodiesterase type 5 (PDE5), which is the responsible enzyme for the degradation of cGMP in the corpus cavernosum [4,5]. SC is used daily with a maximum dose of 100 mg for all categories of patients’ inabilities, whatever the causes of their erectile dysfunction. The elderly, those with hepatic or renal impairments, and/or those receiving cytochrome P450 enzyme CYP3A4 inhibitors could be treated with SC as well [6]. Moreover, the efficacy of SC is a dose-related improvement in the form of its biological duration of erections and hardness and the frequency of the patient’s abilities.

Different developed analytical methods were described for the determination of SC in its oral pharmaceutical tablets, namely high-performance liquid chromatographic (HPLC) [7,8,9,10,11,12,13], spectrophotometric [2,14,15], and electroanalytical techniques (voltammetry) [16,17] and potentiometry [18].

The increasing use of ion sensors and their excellent efficiency and reliability in the medical analysis is putting increasing pressure on analytical chemists to develop new sensors for fast, accurate, reproducible, and selective determinations of various species. Furthermore, a dependable and specific assay is critical for determining the disposition, tolerance, and safety of a drug. Potentiometry with ion-selective sensors is frequently used in the field of pharmaceutical and biological analysis due to its convenience and good sensitivity, which encouraged analytical researchers to develop modern selective sensors for fast, accurate, reproducible, and selective determinations of various drug species [18,19].

Pencil graphite electrodes are considered a versatile form of ion-selective electrodes, characterized by their small size, which results from the absence of the internal filling solution, a rapid response time, and a long lifetime [20,21]. These features enable them to be used in biological systems’ determinations, with no need for further sample pretreatment steps, in the form of extraction or filtration because of their ability to be used with turbid or colored solutions [20,21,22,23,24].

The objective of the present work is to develop a new, accurate, sensitive, time, and duration cost-saving potentiometric method for the determination of SC. The methods utilized the convenient pencil graphite two sensor electrodes depending on the difference in the active pH range for each sensor. Initially, we developed and validated the potentiometric method based on ion-pair complexometric reactions between the target analyte, SC, and two different complexing agents, namely sodium tetraphenylborate (Na-TPB) and/or phosphotungstic acid (PTA), as the electroactive phases. Second, preprepared plasticized poly(vinyl chloride, PVC) membranes were coated into graphite rods and were highly sensitive, selective, reproducible, and accurate toward any of the resulting reaction mixtures of SC-TPB and/or SC-PTA in their solid states. Furthermore, O-nitrophenyl octyl ether (O-NPOE) is classically used as a plasticizer with PVC; many reports have highlighted that dioctyl phthalate (DOP) showed good results with PVC, especially if used in pencil graphite support electrodes [20,24,25,26]. Thus, both O-NPOE and DOP were tested in our study. Moreover, the present method was conducted to determine the SC in its oral tablets and spikes in human urine.

## 2. Materials and Methods

### 2.1. Apparatus

The potentiometric measurements were made at 25 ± 5 °C using a Hanna microprocessor ion analyzer pH/mV meter (model 8417) with the proposed SC-TPB or SC-PTA graphite membrane sensor in conjunction with a double-junction Ag/AgCl reference electrode (Orion 900200), dimensions: 110 × 12 mm (cap dia 16 mm) and containing 10% KCl solution in the outer compartment. A circulator thermostat Model C-100 (Cambridge, England), was used to control the temperature.

### 2.2. Materials

Sildenafil citrate (SC, vit. V) was supplied as a gift from Pfizer, Cairo, Egypt (Pfizer Co., Cairo, Egypt, under the authority of Pfizer Inc., New York, NY, USA). Sodium tetraphenylborate (Na-TPB) and tetrahydrofuran (THF) were purchased from Fluka (Buchs, Switzerland). Phosphotungstic acid (PTA), dioctyl phthalate (DOP), *O*-nitrophenyl octyl ether (*O-*NPOE), high molecular mass PVC, and chloranil were obtained from (Sigma Aldrich, St. Louis, MO, USA). All other chemicals used were of analytical grade unless otherwise stated, and doubly-deionized water was used throughout.

### 2.3. Pharmaceutical Preparation

Viagra^®^ tablets (Batch No: EU/1/98/077/015, Pfizer Co., Egypt), under the authority of Pfizer INC., USA) were labeled to contain 50 mg of sildenafil citrate/tablet.

### 2.4. Preparation of the Solutions and Solids

#### 2.4.1. Preparation of SC Stock Solution

The stock solution of SC (10^−2^ M) in acetate buffer (0.1 mM and pH = 5) was prepared by dissolving an accurately weighed 0.667 g of the pure powder into a 100 mL volumetric flask containing an amount of acetate buffer. After being mixed well, the contents were completed into the mark with the same solvent. Thus, the working solutions of the desired concentrations were prepared by serial dilution in the same solvent.

#### 2.4.2. Preparation of Na-TPB and PTA Standard Solutions

A 10^−2^ M solution of Na-TPB or PTA was prepared by dissolving the accurately weighed amounts into a small amount of double distilled water and then made up to the mark of 100 mL of the volumetric flask by the same solvent.

#### 2.4.3. Preparation of the Expected Interfering Ions Solution

Solutions (10^−3^ M) of the standard interferants such as NaCl, KCl, CaCl_2_, NH_4_Cl, ascorbic acid, glycine, glucose, lactose, fructose, maltose, starch, sucrose, *p*-aminophenol, ephedrine HCl, and *p*-aminobenzoic acid were prepared by dissolving the appropriate masses into the 100 mL portion of double distilled water.

#### 2.4.4. Preparation of the Form of the Ion-Pair Complexes

SC-ion pairs using Na-TPB (SC-TPB) or PTA (SC-PTA) were prepared by mixing 50 mL of the target analyte (10^−2^ M) with 50 mL of any of the Na-TPB and/or PTA reagents, both in a concentration of 10^−2^ M. The formed complexes’ precipitates were filtered off, thoroughly washed with distilled water, and allowed to dry in the open air.

#### 2.4.5. Preparation of Ion-Selective Membranes

The ion-selective membranes (sensor electrodes) selective to any of Na-TPB and/or PTA, as an electroactive phase, were prepared by mixing 190.0 mg of PVC with 350.0 mg of any of the plasticizer of DOP or *O*-NPOE into 5 mL of THF. Thus, the component of each mixture was mix-vortex until obtaining the homogeneous mixture, which then was evaporated at room temperature until a concentrated mixture was obtained. Then it was transferred into a small tube (3 mL in volume). Six graphite rods (3 mm diameter and 10 cm long), prepared from spectroscopic grade graphite, were used as a conducting substrate [23] and were dipped separately into the obtained membrane-coating mixture; then, the rest of the THF solvent was left to evaporate. Thus, thin membranes were formed on the graphite surfaces, and this step was repeated several trials until a suitable membrane with a thickness of 0.2 cm was obtained. Finally, the prepared sensor electrodes were conditioned by soaking them into the SC standard solution (10^−2^ M) for 2 h just before instrumental measurements and were stored in the same solution when not in use.

### 2.5. Analytical Procedures

#### 2.5.1. Sensors’ Calibration Study

Each of the prepared graphite sensor electrodes was calibrated by immersing it in conjunction with the reference electrode in a 50 mL beaker containing 10 mL of SC solution (10^−2^ M) in a concentration ranging from 1.0 × 10^−6^ to 1.0 × 10^−2^ M (*n* = 5) at a pH of 5 using acetate buffer (as mentioned above) with a continues stirring. Then the potential readings were recorded after stabilization to ±0.5 mV. The calibration curves for both ion-pair complexing agents were constructed by plotting the recorded potential versus the logarithmic SC concentration.

It is noteworthy that the sensor was soaked in 10^−2^ M SC solution for 2 h before and stored in the same solution after ending the measurement steps.

#### 2.5.2. Selectivity of the Prepared Sensors Study

The selectivity coefficients $K_{\mathrm{SC},\;B}^{\mathrm{potentiometry}}$ of the SC-TPB and SC-PTA ion-selective sensor electrodes (ISEs) were determined employing separate solution methods (SSM) [18,19,20,21,22,23]. Aliquots (10 mL) of 1.0 × 10^−2^ M SC solution were adjusted to pH 5.0 with acetate buffer, and the SC-TPB or SC-PTA sensors were immersed in the test solution, and the potential was measured. The potentials of 1.0 × 10^−3^ M solutions of the interferents adjusted to pH 5.0 were measured. The selectivity coefficients $K_{D,\;B}^{\mathrm{Potentiometry}}$ were measured by employing a separate solution method (SSM) with the rearranged Nicolsky equation [18,23,24]:(1)$$\log K_{D,\;B}^{\mathrm{Potentiometry}}=\left(\frac{E_{1}-{\;E}_{2}}{S}\right)+\left(1+\frac{z_{1}}{z_{2}}\right)\times \log a$$
where E_1_ and E_2_ are the measured potentials of 1.0 × 10^−2^ M SC alone and in the presence of 1.0 × 10^−3^ M of the interfering substances, respectively, z_1_ and z_2_ are the charges of the SC and interfering species (B). S and A are the slope and intercept of the electrode calibration plot, respectively.

#### 2.5.3. Determination of SC in Viagra^®^ Tablets

Three tablets of Viagra^®^, 50 mg, were pulverized and homogenized carefully. A weight that is equivalent to the average weight of one tablet was dissolved in double distilled water by shaking for 30 min with a mechanical shaker. After filtration, the filtrate was made up to 50 mL in a volumetric calibrated flask. Next, the pH of the solution was adjusted to pH 5, vortex-mixed, and finally, the potentials of the samples were determined using any of the prepared graphite sensor electrodes in the triplicate measurements (*n* = 3), as described in Section 2.5.1.

#### 2.5.4. Spectrophotometric Method (Reported) for Determination of SC

An aliquot of SC solution containing 2.5 mg was mixed with 7 mL of chloranil (5 × 10^−3^ M) in a 10 mL volumetric flask. The content of the flask was completed with acetonitrile and mixed. Then the absorbance of the resulting violet color was measured at 548 nm against an experimental blank under the same conditions [15].

#### 2.5.5. Determination of SC by a Potentiometric Titration

To test the simple, practical applicability of the prepared graphite-O-NPOE membrane of SC-TPB and SC-PTA, sensor electrodes were used as indicator electrodes for the titration of 5.0 mL of 1.0 × 10^−2^ M of SC solution with 1.0 × 10^−2^ M of Na-TPB solution.

#### 2.5.6. Analysis Procedure of Spiked Human Urine Samples

The prepared graphite sensor electrodes were used for the determination of the standard SC in urine samples (*n* = 5) using a standard addition method to conduct the reliability and the required bio-sensitivity. Standard SC was used to spike urine samples (2.0 mL of various working solutions). Then, the developed method in Section 2.5.1 was carried out against a urine blank.

#### 2.5.7. Standard Addition Method

Virtually, the standard addition (or spiking) step was developed by adding a different known volume of a known concentration test solution (1.0 × 10^−2^ M, 1.0–10.0 mL) of SC at pH 5 into a 25 mL measuring flask. Next, the potentials displayed by this test solution were measured before and after the addition of a fixed 1.0 mL aliquot of standard SC solution (1.0 × 10^−2^ M). The change in the electrode potential (ΔE, mV) was then recorded and used for determining SC.

## 3. Results

### 3.1. The Conduction Mechanism of the Graphite-Coated Sensor

With the aid of the modern fabricated graphite-coated sensor electrode, the internal boundary potential (graphite membrane) is dependent on the compound species/concentration present on the graphite-coated surfaces. The cationic conduction (Equations (2) and (3), Figure 1) in the sensor could be attributed to the combination of the membrane TPB or PTA with SC ions in its solution as in the following [23,24]:SC + Na-TPB → SC-TPB(2)
SC + PTA → SC-PTA(3)
where Na-TPB or PTA is embedded in the membrane, and SC-TPB or SC-PTA is the reaction product at the membrane surface. Therefore, the charge transfer across the membrane is carried out by ions coupled to the electronic charge at the graphite surface. Thus, the accumulation of SC ions at the membrane surface will alter its potential.

### 3.2. Sensor Electrode Behavior Study

The general working characteristics of the electrodes were evaluated by performing regular calibrations in SC solutions, as in Figure 2. Moreover, Table 1 shows the performed preliminary studies on the influence of a plasticizer choice as it affects electrode performances. However, the results of the determinations of the SC-TPB or SC-PTA plasticized with *O*-NPOE were compared with those obtained in the case of DOP, and their performance characteristics were evaluated according to the IUPAC recommendations [23,24,25,26]. Of the two tested plasticizers, *O*-NPOE shows the highest total potential change. These obtained data were attributed to the high dielectric constant of *O*-NPOE, and the high extractability of the formed SC-TPB and/or SC-PTA ion-pair complexes into the sensor matrix compared with another plasticizer (ε values are 24 and 5.2 of *O*-NPOE and DOP, respectively) [27]. Therefore, PVC plasticized with *O*-NPOE was used for the next experimental studies.

### 3.3. Effect of pH

The effect of pH on the SC sensor electrodes containing the SC-TPB and SC-PAT with PVC plasticized with *O-*NPOE was investigated by measuring the response to 1 × 10^−3^, 1 × 10^−4^, and 1 × 10^−5^ M of SC solutions at pHs that ranged from 1.0–10. The pH is adjusted using HCl and /or NaOH. Figure 3A,B show that the two sensor electrodes exhibited a stable response over the pH range of 3–6 and 3.3–6, respectively, for SC-TPB and SC-PTA sensor electrodes. At lower pH values, the SC contains more than one positive site (nitrogen centers), which increases the potential response and controls the membrane response and stability. However, at the higher pH values, the potential readings for both sensors decreased sharply due to the precipitation or hydrolysis of the measured SC samples. Thus, if the electrode was applied for the determination of SC in samples with a pH higher than 6, the pH of the sample should be adjusted using HCl.

### 3.4. Response Time Study

The response time was studied by measuring the steady state potential of 10^−4^, 10^−3^, and 10^−2^ M solutions of SC for 3 min. As can be seen in Figure 3C, less than 30 s was the average dynamic response time required to reach the maximum response for SC-TPB sensor electrodes with PVC plasticized with *O-*NPOE. Moreover, it is noted that the response time is more rapid when the measured solutions are lower in concentration; however, 30 s is still optimum for all the measured SC solutions.

### 3.5. Effect of the Diverse (Interfering Ions) Ions on the SC Sensor Electrode

The ions and substances that might exist in biological fluids or SC pharmaceutics, including ions, sugars, amino acids, and different cations, were all tested as possible interferents, and the results are listed in Table 2.

A reasonably accurate selectivity toward SC in the presence of the common nitrogenous compounds such as amino acids and amines and some inorganic cations was observed. The results confirmed that there was no serious interference by various pharmaceutical excipients, diluents, and active ingredients commonly used in the drug tablet formulations, namely lactose, maltose, glucose, starch, talc powder, mannitol, and magnesium stearate, at a concentration as high as a 10–fold molar excess over SC, Table 2.

The other advantage that could be added to the present work is that the plasticized PVC membrane sensors behave in different ways compared to the liquid membrane sensors. The ion exchange sites are poorly mobile, and the coefficients $K_{D,\;B}^{\mathrm{Potentiometry}}$ of such a system is given by the equation:(4)$$K_{D,\;B}^{\mathrm{Potentiometry}}=\frac{U_{B\;}\times {\;K}_{B}}{U_{D\;}\times {\;K}_{D}}$$
where U_D_ and U_B_ and K_D_ and K_B_ are the mobilities and molar distribution coefficients of the SC and the interfering species (B) in the membrane phase and between the aqueous phase and the PVC membrane, respectively.

As reported for the PVC surface membranes, the ions’ presence undergoes mobility restriction owing to their complexation with long-chain complexing agents [18,26]. Hence, the partition coefficients of the SC and the existence of the interfering ions between the membrane and the measured aqueous phase are the main reasons for the observed sensor selectivity.

### 3.6. Determination of SC by Potentiometric Titration

The applicability of the present method was checked using a simple potentiometric titration technique. The results obtained (Table 3 and Figure 4) demonstrated that SC reacts with Na-TPB in a molar ratio of 1:1, and the titration curve was symmetrical with a very well-defined potential jump of about 340 mV, which indicates the high sensitivity of the prepared graphite-O-NPOE and graphite-DOP sensor electrodes as compared with the reported results [26]. Moreover, the method showed its suitability in the form of accuracy and precision, as indicated in Table 3.

### 3.7. Direct Determination of SC in the Viagra^®^ Tablets

The prepared graphite sensor electrode showed acceptable working and indicator characteristics, suitable for SC potentiometric determination of SC in its pharmaceutics. Table 4 summarizes the results of the potentiometric analysis of SC in its Viagra tablets. The results of the proposed methods were compared with those obtained using the reference USP method [1]. The average recovery was 101.34 ± 0.65% with a relative standard deviation (RSD) of ±0.70% for the proposed sensor SC-TPB and 95.26 ± 0.51% and RSD of 1.07% for the proposed SC-PTA. The values presented in Table 4 reveal a good agreement between both methods, the proposed potentiometric and the spectrophotometric recommended by USP [2,15], and there was no difference between the two methods regarding accuracy and precision as revealed by the student’s t-test and the variance ratio F-test.

### 3.8. Analysis SC Spiked in Human Urine Samples

The proposed sensor electrodes were employed successfully for the determination of SC in spiked urine samples. The used urine samples’ pH was measured and was found to be slightly acidic (4–6), which lies in the optimum range of the sensor. Table 5 shows the obtained results, which indicated that the prepared sensor electrodes could detect the SC content in the spiked urine sample with high accuracy and precision. The values presented in Table 5 reveal a good agreement between both methods, the proposed potentiometric and the spectrophotometric recommended by USP [2,15].

### 3.9. Repeatability and Reproducibility Study

The results revealed that there were no changes in the potential response of the two SC-TPB and SC-PAT sensor electrodes over a long time after repeating each measurement five times. The prepared sensors are usable for at least 60 days without any change in their response characteristics after they were stored in 1.0 × 0^−2^ M of SC solution when not in use. The master membrane can be used after four months with the same response when stored below 4 °C, as the results indicated.

### 3.10. Comparison of the Performance of the Proposed Method with the Previously Reported Potentiometric Methods for SC Analysis

The performance of the proposed potentiometric method for the determination of SC was compared with the previously reported electrochemical methods in the literature [18,19,28,29,30,31]. As illustrated in Table 6**,** the current method is more sensitive than methods AdCSV [28] and FIA with an amperometric detector [29] and with a more realistic linear range than the FIA method [18]. Additionally, our method used simpler materials than the Ppy/Cit/Graphite [30] and FIA with amperometric detector [29] methods. Furthermore, the speed here is more rapid than all the previously reported methods, which gives us the advantages of simplicity, rapidity, sensitivity, reproducibility, and accuracy. In conclusion, this method is cheap, fast, and sensitive compared to most of the methods listed in Table 6, as it gives a realistic linear range from 1.0 × 10^−6^–1.0 × 10^−2^ M, and is safe, non-laborious, and not tedious. The sensor used in this method is made of natural graphite, as it is compatible with the application of sustainability standards compared to the other methods referred to in the literature. Furthermore, 95% of the final response was reached within about 15 s compared with the previous methods.

## 4. Conclusions

A rapid and convenient study was utilized in the preparation of modern sensitive, selective graphite-sensitive electrodes used for the first time in the instrumental potentiometry and potentiometric titrimetric determination of SC in pure and pharmaceutical forms as well as in spiked human urine. The method had simple presteps of complex formation between Na-TPB and/or PTA and SC, and then loaded into two types of modern sensor electrodes using PVC with two available plasticizers. Both Na-TPB and PTA fabricated sensors showed similar analytical performance towards SC in terms of linearity and LOD; however, PTA is more cost-effective than Na-TPB; thus, it could be slightly preferred. The profiling of the proposed method was conducted for the determination of SC in its tablets, and urine samples with the % recovery values were compared with those obtained by an official method and showed an RSD ≤ 0.3% (*n* = 5).

## Data Availability

The datasets generated during and/or analyzed during the current study are available from the corresponding author upon reasonable request.

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
