# Peer review of "Two Versatile Pencil Graphite–Polymer Sensor Electrodes Coupled with Potentiometry and Potentiometric Titration Methods: Profiling Determinations of Vitamin V in Tablets and Urine Samples"

_sensors, 2022, doi:10.3390/s22239128_

Round 1
Reviewer 1 Report
1. Rationality behind the choice of plasticizer (DOP and O-NPOE) with PVC is not highlighted in introduction.
2. Authors observed a stable response in the pH range from 3-6 and 3.3-6 for SC-TPB and SC-PTA, respectively. But in general normal urine pH is between 4.6 to 8.0. In such case, what would be the performance of prepared electrode at pH 7 and above?
3. Fig. 3C caption is not clear with the graph. Further the color trace for representing the concentration in all graph should be consistent.
4. Cationic conduction mechanism should be represented in chemical structure format.
5. Out of the prepared ion-pair sensor which composition is exhibiting superior performance SC-TPB or SC-PTA? This information is not clear in the conclusion
6. There are typo errors. Thus, a proof-check is required.
Author Response
Sir, thank you very much.
The response to your valuable and pertinent comments and directions is included in the attached MS doc.

Reviewer 2 Report
1. Please confirm and revise the abstract's representation of concentrations. “1.0 × 10-6 to 1.0 × 10-2 and 1.0 × 10-5 to 1.0 × 10-2 M, with LODs of 6.5 × 10-7 and 5.5 × 10-6” Page 1, line 27-28.
2. In the manuscript, please identify and unify the relationship between unit symbols and numbers, as well as whether spaces are required.
“The % recovery values were compared with 32 those obtained by an official method and showed an RSD ≤ 0.3%” (Page 1, line 33);
“…trode (Orion 90-02) containing 10 %, w/w KCl solution in the outer compartment.” (Page 2, line 86);
“The average recovery was 101.34 ± 0.65 % with a relative standard deviation of (RSD) of ± 0.7 % for the proposed sensor SC-TPB and 95.26 ± 0.53% and RSD for the proposed SC-PTA” (Page 10, line 312-314);
“The tabulated F- and t-values at n = 5 and 95% confidence levels…” (Page 11, line 326);
“…by dissolving an accurately weighed 0.66 g of the pure powder into a 100-mL volumetric…” (Page 3, line 103);
“…conjunction with the reference electrode in a 50-mL beaker containing…” (Page 4, line 138);
“…SC at pH 5 into a 25-mL measuring flask.” and others in the manuscript.
3. Please confirm the correct formulation of the following sentence “…were spiked by a standard SC (2.0 mL of different working solutions ????).”
4. Please confirm the correctness of the article serial number, the main title is “3. Results”, and the following subtitles should be 3.1, 3.2. . . However, the following subtitles is “2.6. The conduction mechanism of the graphite-coated sensor; the same as 2.7-2.14”, which should be rearranged.
5. The expressions in Table 2 should be unified between the multiplication signs and numbers.
6. According to the author, the purpose of the present study is to develop a new potentiometric method for determining SC. The potentiometric method was not clearly described in the introduction, and the advantages of using this method were not clearly stated.
7. In order to illustrate the benefits of the potentiometric method, it would be helpful if the author provided 2-3 examples from domestic and foreign sources.
8. An explanation of the benefits of the potentiometric method in the Analytical procedures was not provided by the authors. Therefore, it is recommended that the author compare the results in this manuscript with those reported in other similar studies.
Author Response

(The authors gave the same response as above.)

Round 2
Reviewer 2 Report
Accept